# Barriers to implementation of enhanced recovery after surgery (ERAS) by a multidisciplinary team in China: a multicentre qualitative study

Dan Wang,[1] Zhenmi Liu,[2] Jing Zhou,[3] Jie Yang,[4] Xinrong Chen,[1] Chengting Chang,[1] Changqing Liu,[1] Ka Li  ,[1] Jiankun Hu[4]

[1]West China School of Nursing/ West China Hospital, Sichuan University, Chengdu, Sichuan, China
[2]West China School of Public Health/West China Fourth Hospital, Sichuan University, Chengdu, Sichuan, China
[3]Hepatobiliary and pancreatic surgery, The Second People's Hospital of Chengdu, Chengdu, Sichuan, China
[4]Department of Gastrointestinal Surgery, West China Hospital, Sichuan University, Chengdu, Sichuan, China

**Correspondence to**
Professor Ka Li;
Lika127@126.com and
Dr Jiankun Hu;
hujkwch@126.com

## ABSTRACT

**Objective** To explore the attitudes and barriers encountered in the implementation of enhanced recovery after surgery (ERAS) in China from the perspective of multidisciplinary team members.

**Design** Based on Donabedian's structure–process–outcome (SPO) model, a multicentre qualitative study using semistructured interviews was conducted.

**Setting** From September 2020 to December 2020, the participants of this study were interviewed from six tertiary hospitals in Sichuan province (n=3), Jiangsu province (n=2) and Guangxi province (n=1) in China.

**Participants** A total of 42 members, including surgeons (n=11), anaesthesiologists (n=10), surgical nurses (n=14) and dietitians(n=7) were interviewed.

**Results** Multidisciplinary team (MDT) members still face many barriers during the process of implementing ERAS. Eight main themes are described around the barriers in the implementation of ERAS. Themes in the structure dimension are: (1) shortage of medical resources, (2) lack of policy support and (3) outdated concepts. Themes in the process dimension are: (1) poor doctor–patient collaboration, (2) poor communication and collaboration among MDT members and (3) lack of individualised management. Themes in the outcome dimension are: (1) low compliance and (2) high medical costs. The current implementation of ERAS is still based on ideas more than reality.

**Conclusions** In general, barriers to ERAS implementation are broad. Identifying key elements of problems in the application and promotion of ERAS from the perspective of the MDT would provide a starting point for future quality improvement of ERAS, enhance the clinical effect of ERAS and increase formalised ERAS utilisation in China.

## Strengths and limitations of this study

► This study is the first multicentre qualitative study of enhanced recovery after surgery (ERAS) in China that explored the barriers encountered in the implementation and promotion of ERAS from the perspective of multidisciplinary team members.

► This study included dietitians as one of the research objects to analyse the views and opinions of dietitians about the implementation of ERAS.

► This study used the structure–process–outcome model as theoretical support to analyse the barriers in the application process of ERAS from a multidimensional perspective.

► Convenience sampling could lead to potential sampling bias.

► There were potential hints and prejudices in questions raised and interviewer's explanations.

## INTRODUCTION

The surgical stress response associated with major surgery describes fundamental metabolic changes that lead to increased catabolism, immunosuppression, free radical production and hypercoagulable states.[1] Enhanced recovery after surgery (ERAS) programmes were introduced and implemented by Danish surgeon Kehlet in the 1990s.[1] The core concept of ERAS is to apply a series of evidence-based, innovative and optimised perioperative measures through multidisciplinary teamwork, such as prehabilitation, minimally invasive surgery, removal of invasive lines, multimodal analgesia, early mobilisation and early oral feeding, to reduce ineffective and unhelpful medical interventions and reduce the incidence of postoperative stress and complications, thereby speeding up the recovery and adaptation of patients after surgery.[2–5] Based on these advantages, the ERAS programme has been adopted internationally and applied widely in the field of surgery.

In China, the ERAS programme was first proposed and implemented by Professor Jieshou Li in many fields of general surgery, such as stomach and colorectal surgery, and achieved a good curative effect for patients.[6 7] Subsequently, ERAS programmes have been popularised and applied in tertiary hospitals (the highest level) in various provinces of

China for more than 10 years. Through a large number of randomised controlled trials, it has been proven that the implementation of ERAS can effectively reduce the pain of patients, hospitalisation costs and length of stay in the hospital without increasing the 30-day readmission rate and complications.[8–11] At present, the ERAS programme has been successfully applied to general,[12] orthopaedic,[10] urological[11] and other operations,[13] and many expert consensuses have been developed for each specialty, marking the maturity of the application of ERAS in China.

There is currently sufficient evidence to support the hypothesis that the implementation of ERAS can indeed effectively improve the efficiency of the use of health service resources[14] and improve patient satisfaction with the diagnosis and treatment results.[15 16] However, there are variations in the implementation of ERAS for different regions, hospitals at different levels and experts in different disciplines,[17–19] which limits the application and popularisation of ERAS.

The ERAS pathway is carried out in the form of multidisciplinary cooperation, including multiple modes of surgery, anaesthesia, nursing and perioperative management to ensure the safety and effectiveness of patients during the perioperative period.[4] Previous studies have proven that multidisciplinary teams (MDTs) are an important guarantee for ERAS, and the effective cooperation of MDTs is a key positive factor in promoting the application and advancement of ERAS.[20] Overall, the establishment of a patient-centred MDT is an inevitable trend in the development of ERAS. However, the construction and management of ERAS–MDT is still in the exploratory stage, and multidisciplinary members still face many barriers and challenges in the various processes of implementing ERAS.[21 22] Therefore, a targeted analysis of various barriers and a quality management plan are crucial to the construction and development of ERAS–MDT, which is expected to speed up the implementation of ERAS in China.

Previous studies have found many barriers to interprofessional cooperation among ERAS–MDTs, such as a lack of personnel, resistance to changes in traditional surgical concepts, poor patient compliance and collaboration, and a lack of unified management and operative plans for MDTs.[22–25] However, the current qualitative research mainly focused on surgeons, nurses and anaesthesiologists.[25] There are only limited data on the barriers encountered by dietitians in the implementation of ERAS,[26] warranting further study.

In this study, based on Donabedian's structure–process–outcome healthcare quality model (SPO model), we presented the results of a qualitative study that aimed to gain an in-depth understanding of the views and opinions of multidisciplinary members in different regions on the implementation and promotion of the ERAS programme and to explore the barriers that affect the effective implementation of ERAS, so as to provide a basis for the establishment of the quality evaluation system of ERAS, which can contribute to the development of ERAS programmes in China.

## METHODS

Based on the SPO model, we conducted a multicentre qualitative interview study to explore the experience of surgeons, anaesthesiologists, nurses and dietitians in implementing the ERAS programme from the three aspects of structure, process and outcome and to analyse the barriers that affect the promotion and application of ERAS programmes.

### Theoretical framework

The SPO model proposed by Donabedian considers both internal and external organisational factors. Internally, it covers an organisation's structure (S), process (P) and outcome (O), and their interactions. On the other hand, external factors, such as health and social factors, are also combined in this model.[27 28] The SPO model also has theoretical underpinnings in that good structure should promote good process and good process should promote good outcomes, which is conducive to guiding the effective improvement of medical quality.[29] Therefore, the approach of Donabedian is valuable in conceptualising and organising medical quality evaluation,[30] and has been widely adopted in the research of ERAS quality evaluation.[31] In this qualitative interview study, the SPO model was used to analyse the barriers of SPO that potentially affect the implementation of ERAS programmes. The 'structure' refers to the methods of resource allocation and the efficiency of resource utilisation in medical services, such as the adequacy of facilities and equipment, sufficient staff and financial resources, and complete management policies. 'Process' refers to the specific execution quantity and quality of each link in the implementation of medical activities according to the plan, such as the execution rate and execution quality of each ERAS programme. The 'outcome' refers to the final result after the end of medical services, such as patient satisfaction, rehabilitation effect and medical cost.[27 28]

### Study design and setting

Data collection used a qualitative descriptive approach and semistructured interviews. The ERAS programme was mainly implemented in tertiary public hospitals in China, and convenience sampling was used to select hospitals in our study. The criteria for selecting the hospitals are shown in figure 1. Participants in our study were interviewed from six tertiary hospitals in Sichuan province (n=3), Jiangsu province (n=2) and Guangxi province (n=1). The ERAS implementation process, year, surgical field, compliance and institutional protocol of the hospitals were shown in online supplemental file 1. We also signed the multicentre institutional protocol, and the institutional protocol was the Chinese version (online supplemental file 2).

### Sample

Purposive sampling was used to provide a diverse range of ages, genders, occupational classification and locations (table 1).[32] Overall, 42 members were included in

- ➢ A tertiary public hospital.
- ➢ A general hospital or a specialist hospital.
- ➢ The hospital has been implementing an ERAS program for at least 5 years.
- ➢ The hospital agreed to participate in this study.
- ➢ According to the principle of data saturation, sampling was stopped when there were no new codes or themes.

**Figure 1** A summary of the criteria for selecting hospitals. ERAS, enhanced recovery after surgery.

the analysis, including 11 surgeons, 10 anaesthesiologists, 14 surgical nurses and 7 dietitians; 9 were excluded: 2 for poor quality of recording, 2 for schedule clash, 5 for data saturation. The inclusion criteria of the ERAS–MDT members were as follows: (1) experience in ERAS implementation; (2) more than 10 years of working experience; and (3) gave informed consent and volunteered to participate in the study. Participants who could not express their ideas accurately and fully were excluded.

According to the inclusion and exclusion criteria, the relevant persons in charge of each hospital selected the research subjects who met the purpose of this study and sent a message to invite these research subjects to participate in this study. If they agreed to participate and share their experience and views on the implementation of ERAS, the first author (DW) contacted them by telephone and registered them to participate in this study with written informed consent.

### Data collection

Interviewees were categorised by location of work site (ie, Sichuan, Jiangsu or Guangxi Province) and role (ie, surgeon, anaesthesiologists, surgical nurse and dietitian). The participants were interviewed individually in the period from September to December 2020. Participants

**Table 1** Characteristics of interviewees

| | |
|---|---|
| Mean age (SD, range) | 37.36 (SD 5.94; range 28–52) |
| Characteristic | n (%) |
| Gender | |
| Male | 20 (47.62) |
| Female | 22 (52.38) |
| Occupational classification | |
| Surgeon | 11 (26.19) |
| Anaesthesiologist | 10 (23.81) |
| Surgical nurse | 14 (33.33) |
| Dietitian | 7 (16.67) |
| Location | |
| Sichuan | 25 (59.52) |
| Jiangsu | 10 (23.81) |
| Guangxi | 7 (16.67) |

in Chengdu, Sichuan province, conducted face-to-face interviews in a quiet room at their hospitals. Participants from other provinces were interviewed by telephone. The principal investigator (DW) and research assistant (JZ, XC and CC) conducted the interviews together. All interviewers have completed training in interview. None of the participants had a close personal relationship with the interviewers, and all participants agreed to be audiotaped. According to the SPO model, a semistructured topic guide was used for the interviews, as follows:

- ▶ Judging from the current hospital policy management and basic resources, what do you think are the key factors that hinder the implementation of ERAS?
- ▶ During the perioperative period, which measures recommended by the ERAS guidelines have barriers and difficulties in implementation? What factors caused these barriers?
- ▶ What difficulties are faced by MDTs in the process of cooperating to implement ERAS programmes?
- ▶ What barriers do patients and their families cause to the implementation of ERAS programmes?
- ▶ From your professional perspective, do you think what measures should be taken to improve the implementation effect of ERAS programmes?

During the interview, we flexibly asked different questions based on the respondent's answers and the semistructured nature of the interviews allowed participants to explore other topics which they considered relevant. Meanwhile, interviewers promptly confirmed the vague information provided by the interviewees. The average interview time was 21.84 min. At the end of interviews, the researcher made a brief summary to determine whether there was any missing or additional information and asked the interviewees to fill in the general situation form. The researcher undertook reflection after each interview to check whether the interview process was inappropriate or needed to be improved and to confirm whether the data had reached saturation and whether the interview process needed to continue. This work adheres to Standards for Reporting Qualitative Research.

### Data analysis

The recorded interview was transcribed verbatim within 24 hours by member of the research team. Transcribed scripts were assigned a unique code in the order of interviews, and names were removed to align with confidentiality. Data analysis followed a thematic approach of induction and explanation,[33] applying principles of constant comparison to analyse differences across cases.[34] Main concepts and themes within the data were identified through a combination of open coding and thematic analysis.[35 36] The research team repeated line-by-line reading of the transcripts. Following familiarisation, the research team manually did open coding, drawing on their rich clinical experiences and recorded these codes on the margin of the printed transcript.[37] These codes were subsequently grouped according to the SPO model. Through clustering and integration of codes,

themes of each dimension were determined. Themes were considered for inclusion in this report if they were prominent. Prominent themes were further explored in context through in-depth reading and refined until saturation was reached.[38] To ensure reliability, the research team met regularly to review the coded data, verify its relevance to main themes, discuss the interpretations and agree on any new theme which were required. Data collection ceased when no new codes or themes were identified, which meant data saturation. Data saturation was considered to be the point at which coded data from new interviews were only added to existing themes and no new themes were developed.[39] This study finally identified eight themes.

### Patient and public involvement

No patients were involved in this study. The public were not involved in the development of the research questions, research design or outcome measures.

## RESULTS
### Structure
#### Shortage of medical resources

All participants described the shortage of medical resources as the main barrier to their implementation of the ERAS programme, and the most common description was the pressure and challenges caused by the shortage of human resources. Due to the application of ERAS, many measures needed to be implemented, and to shorten the average length of stay and improve the bed turnover rate, doctors and nurses must take care of more patients with limited resources and time (table 2, Q1–Q2). Many participants described that the workload of doctors and nurses has increased, but hospital leaders have not increased their corresponding salaries and bonuses, which could not motivate the medical staff to implement the ERAS programme (table 2, Q3). Surgeons and anaesthesiologists in three hospitals all mentioned that the operating room of the hospital was not sufficient to provide more equipment and space to satisfy all patients who could carry out the ERAS programme (table 2, Q4–Q5).

#### Lack of policy support

At the hospital level, many hospitals have not formulated and introduced corresponding incentive policies and measures to guide department leaders and medical staff to actively implement ERAS, so many medical staff do not generate internal driving forces and are unwilling to implement ERAS programmes (table 2, Q6). Some participants believed that the ERAS programme promoted by tertiary hospitals is generally not implemented in basic-level hospitals, and the disconnection of concepts cannot ensure the smooth implementation of ERAS in the entire treatment process of patients (table 2, Q7). In addition, there is no formal training course or mechanism for the ERAS programme in China. Medical staff only learn ERAS through meetings, discussions in the department and exchanges with other hospitals. This informal training

**Table 2** Illustrative quotes representative of theme 1, theme 2 and theme 3

| Q | Illustrative quotes |
|---|---|
| 1 | Since the number of nurses did not increased, the quality of implementation would be affected when the workload was particularly heavy. (Interview 39, nurse, female) |
| 2 | …some surgeons thought that getting a nerve block would delay his operation time… Because there were too many patients in China, surgeons had to perform more operations in a limited time… (Interview 13, anaesthesia, male) |
| 3 | …our salary and bonuses did not change after implementing so many ERAS measures. (Interview 25, dietician, male) |
| 4 | Not every patient in our department routinely uses equipment such as insulation blankets and infusion fluid warming, and not every patient can be monitored for temperature. (Interview 14, anaesthesia, male) |
| 5 | If we have more operating rooms, our nerve block can be performed in the preparation room in advance, so the whole process will be smoother. (Interview 19, anaesthesia, female) |
| 6 | Er, I think hospital leaders can stimulate our enthusiasm for implementing ERAS through various policies. For example, … the clinical implementation effects and scientific research results of ERAS should be vigorously publicized. (Interview 11, anaesthesia, male) |
| 7 | Currently, basic-level hospitals generally do not carry out ERAS programs. This is actually a big problem. (Interview 2, nurse, female) |
| 8 | There is currently no standardized ERAS training. If this can be realized in the future, ERAS can be implemented in a more unified and standardized way… (Interview 35, anaesthesia, male) |
| 9 | Many surgeons are based on their own practice standards and are reluctant to accept the various new concepts of ERAS. This is the most critical issue… (Interview 11, anaesthesia, male) |
| 10 | We require local anaesthesia and analgesia for wounds, but surgeons believe that this will lead to liquefaction of the wound and increase the incidence of infection. Although we told them that this will not increase the incidence of wound infection, they are very stubborn, and it is difficult for us to communicate with them. (Interview 14, anaesthesia, male) |

ERAS, enhanced recovery after surgery.

**Table 3** Illustrative quotes representative of theme 4

| Programmes | Q | Illustrative quotes |
|---|---|---|
| Preoperative education | 1 | Patients often complained that doctor were too busy, and they could not find doctors if they wanted to consult. So have patients been given enough information by staff in hospitals? (Interview 32, surgeon, male) |
| | 2 | Sometimes we thought we had explained contents clearly to the patient, but in fact he might not truly understand them. (Interview 27, nurse, female) |
| Preoperative fasting | 3 | Patients with traditional surgery experience would worry about aspiration, so he would not eat anything. Some patients would forget the exact time of fasting or postpone the prescribed time. (Interview 40, dietician, female) |
| Early oral feeding | 4 | Patients would not voluntarily go to eat early after surgery, so we must always supervise and remind them. Some patients were reluctant to eat carbohydrate nutrient solution, mainly because they were worried that the gastrointestinal tract would be uncomfortable after taking it. (Interview 6, nurse, female) |
| | 5 | We encouraged patients to eat early after surgery, but if they had gastric bloating after eating, they would question our ERAS. (Interview 30, surgeon, male) |
| Early mobilisation | 6 | The traditional Chinese concept is to lie in bed for a long time after surgery, which is difficult for patients to change. (Interview 20, anaesthesia, male) |
| | 7 | We would tell patients who early mobilization could enhanced recovery, but they would question this measure, worrying whether it would make surgical wound dehiscence or exacerbate pain. (Interview 1, nurse, female) |
| Postoperative follow-up | 8 | We hope to get the evaluation of the nutritional management effect of patients throughout the perioperative period. However, for example, patients with gallstone surgery were only hospitalized for one day, which made it difficult for us to obtain objective indicators for postoperative nutritional assessment. (Interview 23, dietician, female) |
| | 9 | The medical care and security that patients received after discharge were not enough. Once complications occurred and patients were re-admitted to the hospital, the doctor-patient conflict would become more intense, which would cause many mental pressure on doctors. (Interview 28, surgeon, male) |

ERAS, enhanced recovery after surgery.

does not have a unified ERAS implementation process, which leads to great differences in the implementation of ERAS in various hospitals, and it is impossible to compare and analyse the implementation effects of multicentre hospitals (table 2, Q8).

### Outdated concepts

Doctors are reluctant to try new concepts mainly because they are worried about the safety of patients. The doctor–patient relationship is more subtle in China. If doctors encounter problems in the process of implementing ERAS, they will face greater work pressure and professional crises. Doctors take patient safety as their first priority in any medical measures they take, so they dare not try ERAS lightly (table 2, Q9–Q10).

### Process
### Poor doctor–patient collaboration

Participants also mentioned the importance of doctor–patient collaboration and described situations that hinder the development of some ERAS programmes due to the unsatisfactory process of doctor–patient collaboration (table 3). In terms of preoperative education, doctors believed that because patients have a low degree of understanding and significant medical knowledge gaps, if the information conveyed to patients is not sufficient and clear, the patient will not be able to accurately understand the specific connotations of the doctor's instructions, which will lead to deviation in the process of cooperation with patients and hinder the normal development of some ERAS programmes (table 3, Q1–Q2).

The medical staff mainly mentioned that patients would mistakenly remember the time of preoperative fasting, which would affect ERAS implementation and delay the operation time. Some patients who have undergone traditional surgery will question the ERAS concept of preoperative fasting, worrying about accidents during the operation (table 3, Q3). In terms of early oral feeding, the nurses mentioned that they must personally supervise the patients to make them eat on time. If the supervision is not timely, the patient will delay eating. Some patients will refuse to eat early if they feel unwell, or if they feel unwell after eating, they will be dissatisfied with the quality of ERAS (table 3, Q4–Q5).

Participants said that the traditional concept of patients was that they needed to lie in bed for a long time after surgery because they worried that early mobilisation would affect wound healing and aggravate pain (table 3, Q6–Q7). All dietitians mentioned that due to the shortened length of hospital stay and the lack of a complete postoperative follow-up mechanism, they could only obtain some subjective indicators through the phone, which made them unable to objectively and accurately evaluate the implementation effect of the ERAS nutrition programme. The doctors believed that due to the shortened length of hospital stay, the patient's later treatment cannot be effectively guaranteed, which would increase the dissatisfaction of the patient. If postoperative complications occurred and the patient was re-admitted to the hospital for treatment, it would intensify the doctor–patient contradiction (table 3, Q8–Q9).

Participants emphasised that patients' trust and adequate education were the key factors to improve the effect of doctor–patient collaboration. Some staff believed that patients' trust in doctors' diagnostic and treatment skills was the cornerstone of the implementation of ERAS. In addition, adequate education could

| Q | Illustrative quotes |
|---|---|
| | **Table 4** Illustrative quotes representative of theme 4, theme 5 and theme 6 |
| 1 | My personal view is that the patient needs to trust 100% of the diagnosis and treatment made by the medical staff. The second is that medical staff need to do adequate education with patients, such as telling patients about the benefits that new concepts and measures can bring to him, encouraging him to accept changes, and better cooperating with us in various tasks. (Interview 6, nurse, female) |
| 2 | In fact, hospital leaders have been talking about strengthening multidisciplinary cooperation, but now there are very few that truly realize multidisciplinary cooperation, and for teams that do well in multidisciplinary cooperation, they have not been encouraged and played a leading role. (Interview 14, anaesthesia, male) |
| 3 | … multidisciplinary cooperation is the greatest difficulty in promoting ERAS. (Interview 25, dietician, male) |
| 4 | Um… I think we should reach a consensus on how to carry out multidisciplinary collaboration in ERAS, and formulate a corresponding process. For example, the division of labour in each discipline is clear, and there is a dedicated person to coordinate the work between multiple disciplines. (Interview 22, dietician, female) |
| 5 | Current consensuses of experts were standardized procedures applicable to the general public, there were still individual differences between patients. (Interview 12, anaesthesia, male) |
| 6 | In fact, some patients could not perform activities so early after surgery. For example, for elderly patients, the time when they could perform activities may be relatively late. We still have to apply this concept of ERAS in an individualized way. (Interview 23, dietician, female) |

ERAS, enhanced recovery after surgery.

promote patients' understanding of the diagnosis and treatment plan and further improve patient satisfaction and compliance (table 4, Q1).

### Poor communication and collaboration among multidisciplinary members

Participants mentioned that the main difficulty affecting the implementation of the ERAS programme was the communication and collaboration between MDTs, which would not only have a negative impact on the treatment of patients but also bring challenges to the smooth implementation of ERAS (table 4, Q2–Q3). Participants described that the continuous and effective communication and collaboration of the MDT throughout the perioperative period played a vital role in the smooth progress of ERAS. They also emphasised that different disciplines must reach a consensus on the ERAS programme and jointly discuss and formulate a series of standardised implementation standards and procedures to improve the overall efficiency of collaboration between ERAS–MDT and increase the success rate of ERAS implementation (table 4, Q4).

6.Lack of individualised management

Surgeons and anaesthesiologists mentioned that there was a general lack of personalised perioperative management programmes for ERAS. Due to great individual differences of patients, medical staff should fully consider the individual characteristics of each patient's condition when implementing the ERAS programme and adopt individualised management plans. Otherwise, safety problems may occur for some patients with a complicated aetiology and many serious comorbidities, which will affect the implementation effect of ERAS (table 4, Q5–Q6).

### Outcome
### Low compliance

All participants believed that the compliance of patients and families was one of the important factors affecting the effectiveness of ERAS and that good compliance was an important prerequisite for ensuring the safety and effectiveness of treatment. Participants mentioned that the poor compliance of some patients and their families would reduce the effectiveness of ERAS and even cause a waste of medical resources (table 5, Q1–Q2).

### High cost

Anaesthetists mentioned that to save money, patients with poor economic conditions would choose to endure pain and refuse to use nerve blocks or analgesic pumps. Nutritionists believed that due to the high price of commercial nutritional preparations and the fact that medical insurance does not reimburse this part of the cost, some patients would refuse to use nutritional preparations, leading to the lack of essential nutrients in the diet, which affects the postoperative nutritional treatment (table 5, Q3–Q4).

### DISCUSSION

Based on SPO theory, we conducted a qualitative study on multidisciplinary ERAS members of six tertiary hospitals in three provinces in China. We coded and refined the theme from the three dimensions of SPO and summarised eight main barriers. To our knowledge, this is the first multicentre qualitative article that analyses ERAS application barriers from the perspective of multidisciplinary members in China.

Although medical staff implement the ERAS programmes under the guidance of the ERAS guidelines,

**Table 5** Illustrative quotes representative of theme 7 and theme 8

| Q | Illustrative quotes |
|---|---|
| 1 | I Occasionally, patients thought that the pain could be tolerated, so they arbitrarily refused to take analgesics prescribed by doctors. However, their pain would increase during activities, which would inevitably affect the implementation of early mobilization and other programs. (Interview 7, nurse, female) |
| 2 | Some patients were unwilling to be discharged early for fear that treatment after discharge would not be guaranteed… patients and families might lie about their illness, which greatly affects the doctor's diagnosis… This way would lead to prolonged hospitalization and waste of resources. (Interview 8, surgeon, male) |
| 3 | … the cost would increase by approximately 500 yuan… those patients with poor financial conditions would refuse to do the nerve block. In addition, the cost of postoperative analgesia pumps was nearly 1000 yuan, and there were some patients who would rather bear the pain than spend more money. (Interview 19, anaesthesia, male) |
| 4 | The price of nutritional preparations is very expensive, with a bottle cost of more than 80 yuan. Patients needed to use at least 5 bottles before and after surgery, which would cost more than 400 yuan, and medical insurance was not reimbursed for this cost. (Interview 21, dietician, male) |

we found that the audit system has not been established or perfected, many feedback problems cannot be effectively resolved, causing MDT members to face many barriers in the process of implementing ERAS. On the structural dimension, limited medical resources were not conducive to improving the quality of medical services of the ERAS programme. Similar to previous evidence,[21 25] we found that in the reality of a heavy disease burden and overloaded work intensity, the shortage of doctors, nurses and other medical staff has led to poor completion quality or failure to implement some ERAS programmes, which has become an important factor hindering the sustained, stable and rapid development of ERAS. New findings in our study were that due to the funding gap between government financial appropriations and the actual needs of hospitals, and the government and hospitals had insufficient special financial support for medical and health resources of ERAS, resulting in insufficient reserves of beds, equipment, medicines and other materials. In addition, MDT members who implemented ERAS did not receive increased salaries or bonuses, which led to the slow application of ERAS in clinical departments and did not promote a large-scale development trend. We recommend that hospital leaders and managers should pay more attention to optimising the complex process of ERAS, reducing unnecessary work, increasing funding for ERAS application and rewarding MDT members who implement ERAS, so as to actively promote the implementation and development of ERAS.

The medical treatment combination was one of China's important measures to deepen the reform of public hospitals, aiming to build a scientific and reasonable medical service system through the alliance and resource sharing of medical and health institutions at different levels.[40] However, we found that the new concepts and new technologies of the ERAS programme in primary hospitals are still in the exploratory stage, failing to keep up with the implementation of ERAS in tertiary hospitals, resulting in patients still receiving traditional treatments during recuperation in primary hospitals after discharge, which makes the application of ERAS appear inconsistent.

In addition, unlike other studies,[22 25] we found that the absence of a unified standardised ERAS course training mechanism in China has led to deviations in the understanding and mastery of ERAS among multidisciplinary members. Previous studies have proven that the establishment of standardised ERAS training courses would provide more scientific and standardised knowledge education and skills training, which would help medical staff to change their concepts and achieve better consistency.[41] We suggested that policies should be issued at the national level to vigorously support and promote the development of ERAS, build and promote a standardised training system for ERAS, improve the awareness and acceptance of ERAS among medical staff at all levels of medical units, update and unify new concepts in a timely manner, promote the construction of the ERAS medical consortium and continue to improve supporting policies to strengthen the service capabilities of medical and health institutions.

In the process dimension, the lack of doctor–patient collaboration was one of the barriers to the implementation of ERAS, which could appear in all aspects of the perioperative period. For example, patients are reluctant to follow the advice given by medical staff to engage in early activities after surgery. The reasons are related to patients' poor understanding and acceptance of ERAS, as well as ineffective communication between doctors and patients. Studies have confirmed that good doctor–patient collaboration is an important factor in promoting the effective, safe and efficient operation of the healthcare system.[42] Our study also showed that patients' high trust in medical staff and adequate education by medical staff are the key factors to improving doctor–patient collaboration. Medical staff need to spend enough time providing diagnostic and treatment services for patients and clearly explain their condition to patients, which is a favourable factor for enhancing the collaboration between staff and patients.[13] In addition, individualised management of patients can improve the safety and effectiveness of the implementation of ERAS, which can alleviate patients' worries about the concept, measures and effects of ERAS

so that they can trust the medical plans adopted by staff and actively participate in their treatment and recovery.

Previous studies have proven that close multidisciplinary cooperation is an inevitable requirement in the development of ERAS.[43] However, similar to previous evidence,[44] our study also demonstrated that the existing ERAS–MDT model still faces many difficulties in clinical practice. The current model lacks a unified and reasonable overall structure and construction plan, and multidisciplinary members have not fully reached a consensus on ERAS. Furthermore, the lack of scientific management of the implementation path and collaborative operation of ERAS–MDT result in the unclear responsibilities and division of labour of ERAS–MDT members. Meanwhile, the communication, coordination and perioperative connection of various links need to be further optimised. Therefore, We suggest that taking the patient as the centre and relying on an MDT to formulate a standardised, individualised, and continuous comprehensive diagnosis and treatment plan is essential for the implementation of ERAS on the process dimension.

On the outcome dimension, participants agreed that poor compliance of patients and families would cause the quality of ERAS implementation to decline. Previous studies have shown that increased compliance with ERAS programmes would lead to better treatment outcomes and may have long-term benefits for survival.[23] Therefore, the compliance of patients and families to all ERAS programmes is an important indicator to ensure the implementation effect of ERAS. Our study demonstrated that high costs were one of the main reasons for the poor compliance of patients and families. With the continuous process of medical reform, the Social Basic Medical Insurance has covered more than 97% of residents in China and has gradually become an important way to solve the problem of expensive medical treatment for residents.[45] However, it is not free medical care and patients still have to pay part of the cost, which would make some patients with poor financial conditions resist some self-financed treatment measures, such as not eating nutritional preparations and refusing analgesic pumps. We suggest that the government and hospitals should actively introduce plans to reduce medical costs and drug expenses, and further expand the items and scope of medical insurance reimbursement, thereby improving the compliance of the ERAS project.

With the SPO model as a theoretical support, this study is an innovative multicentre qualitative study of ERAS in China that explores the barriers encountered in the implementation and promotion of ERAS from the perspective of MDT members. In addition, this study included dietitians as one of the research objects to understand and analyse the views and opinions of dietitians on the implementation of ERAS to compensate for the limitations of previous studies. This study also has limitations. First, the survey results were limited by the self-reporting nature of the interviews. We tried our best to collect different levels of information in terms of age, gender, working years, professional title, and working region to compensate for the limitations of personal cognition. Second, there were potential hints and prejudices in both the questions raised and the guidance and explanations of the interviewer. We selected assistants who did not know ERAS to participate in the interview to avoid personal experience affecting the views of the research subjects. Finally, this study did not interview members of all disciplines involved in the implementation of ERAS, such as pharmacists and rehabilitation doctors. In addition, the attitudes and opinions of health system leaders and patients play a vital role in the application and development of ERAS. In the future, we will further expand the types of research objects and improve the research content.

## Conclusions

This study analysed the barriers to the implementation of the ERAS programme in the SPO dimensions from the perspective of multidisciplinary members. Although a large amount of data has shown the advantages of the ERAS programme, this study reveals the fact that the implementation of ERAS still faces many obstacles, and there is need to enrich supporting resources, optimise processes to improve implementation effects in China. This study is a multicentre qualitative study, and all hospitals included belong to the southern region of China. Therefore, our results can partially reflect the application status of ERAS in the southern region. In general, the current implementation of ERAS was still based on ideas more than reality. This study has identified common barriers for MDT members to use ERAS in the three provinces of China, and have provided pragmatic solutions to each obstacle. More quality improvement research, evaluation and audit will be needed to improve the application effect of ERAS in the future. We hope that this study provides a starting point for future quality improvement of ERAS, enhance the clinical effect of ERAS and increase formalised ERAS utilisation in China.

**Contributors** KL planned the study. DW was principal investigator, collected, analysed and interpreted the data and wrote the manuscript. KL, DW and ZL designed the study, developed the methods. ZL, CL and JH reviewed/edited the manuscript. JZ, CC, XC collected, analysed and checked the data. JY helped draft the paper, provided critical reviews and intellectual content. All authors have seen and approved the final version of the manuscript. DW is the guarantor. KL, ZL and JH are on the supervisory team for DW PhD study.

**Funding** This work was supported by a program of National Natural Science Foundation of China (71974135). The funders had no role in the study design, data collection and analysis, decision to publish, or preparation of the manuscript.

**Competing interests** None declared.

**Patient consent for publication** Not applicable.

**Ethics approval** This study involves human participants and was approved by the Ethics Committee of Biomedical Research (West China Hospital of Sichuan University) (No. 2020-1038). Participants gave informed consent to participate in the study before taking part. Data were treated confidentially. Each participant was assigned a code before data analysis (eg, interview 2, anaesthesia, male), and therefore, personal identification was removed.

**Provenance and peer review** Not commissioned; externally peer reviewed.

**ORCID iD**
Ka Li http://orcid.org/0000-0002-4580-0015

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
