## [Reviewer comments · BMJ Open]

ARTICLE DETAILS

TITLE (PROVISIONAL)	Barriers to implementation of enhanced recovery after surgery (ERAS) by a multidisciplinary team in China: a multicentre qualitative study
AUTHORS	Wang, Dan; Liu, Zhenmi; Zhou, Jing; Yang, Jie; Chen, Xinrong; Chang, Chengting; Liu, Changqing; Li, Ka; Hu, Jiankun

VERSION 1 – REVIEW

REVIEWER	HUBNER, Martin CHUV LAusanne
REVIEW RETURNED	09-Jul-2021

GENERAL COMMENTS	Dr. Wang and colleagues conducted a qualitative study to assess barriers to ERAS implementation in China. Using a SPO model, the interviewed 42 MDT members from 4 different disciplines identifying multiple different limitations for the 3 dimensions of structure, process and outcome. Implementation remains THE challenge for successful ERAS care and the topic is hence of high importance. Several issues need to be addressed by the authors though: 1. The manuscript is overly descriptive and much too long. Despite the qualitative nature, hrouping of answers in form of tables and comparisons between professions would be helpful for understanding, joy of reading and shortening of the manuscript.2. What are next steps? This should be stated in the conclusion which is currently more than vague.3. The Donabedian's SPO model should be briefly explained under methods.4. Key refs on implementation and hurdles for implementation are missing in the ref list and in the discussion.5. Selection of MDT members should be clarified: How many wee eligible according to inclusion criteria in the different institutions, how many were not included (reasons)?6. To what extent can the results be generalized to the world or at least to the rest of China?7. some details on ERAS implementation should be given: implementation process, year of implementation in the centers, fields of surgery implemented; provide institutional protocol as online appendix; provide some data on compliance in the different centers.8. The discussion is full of repetition of results: please moit redundancy but set your results in the context of the literature and draw conclusions how this all will change your pratice or shape new research projects?!9. Please provide tables in the end of the manuscript and not embedded in the text.10. Table 1 could be nicely provided as figure.
---

REVIEWER	Gramlich , Leah University of Alberta, Medicine
REVIEW RETURNED	13-Sep-2021

GENERAL COMMENTS	Thanks for this paper. The rationale for using the donabedian approach - SPO - needs more justification and perhaps an acknowledgement of other potential approaches The lack of description of ERAS program characteristics/expectations within the programs (eg - do they use an evidence based care pathway, do they use implementation teams to review data and practices with change management approaches, do they use an audit system to inform practice change) is a gap The authors do not describe involvement of either patients or health system leaders - this is a gap that could be explained in the discussion The current evidence on barriers and enablers to ERAS care is not completely described based on the literature The process/approach to qualitative evaluation is not well described - eg how and whata themes were identified this needs a significant rewrite but it is important information
---

VERSION 1 – AUTHOR RESPONSE

Reviewer: 1

Dr. Martin HUBNER, CHUV Lausanne

1. The manuscript is overly descriptive and much too long. Despite the qualitative nature, hrouping of answers in form of tables and comparisons between professions would be helpful for understanding, joy of reading and shortening of the manuscript.

Response: Thank you very much for this good suggestion. According to your suggestion, we simplified the content of the results and made tables of all quotes, namely Table 2 to Table 5 of the attached "Main Document-marked copy".

2. What are next steps? This should be stated in the conclusion which is currently more than vague.

Response: Thank you for such detailed advice. According to your suggestion, we described our next plan in detail. They are the highlights in lines 486-490 of the attached "Main Document-marked copy". The specific content is as follows:

More quality improvement research, evaluation and audit will be needed to improve the application effect of ERAS in the future. We hope that this study provides a starting point for future quality

improvement of ERAS, enhance the clinical effect of ERAS and increase formalized ERAS utilization in China.

3. The Donabedian's SPO model should be briefly explained under methods.

Response: Thank you for such detailed advice. According to your suggestion, we have added the content of the theoretical framework to the method. This part of the content introduces in detail what is the SPO model, the specific content of the SPO model, the reason why we choose the SPO model, and how SPO guides our research. They are the highlights in lines 136-147 of the attached "Main Document-marked copy". The specific content is as follows:

The SPO model proposed by Donabedian considers both internal and external organizational factors. Internally, it covers an organization's structure (S), process (P), and outcome (O), and their interactions. On the other hand, external factors, such as health and social factors are also combined in this model.^{Error! Reference source not found. Error! Reference source not found.} The SPO model also has theoretical underpinnings in that good structure should promote good process and good process should promote good outcomes, which is conducive to guiding the effective improvement of medical quality.^{Error! Reference source not found.} Therefore, the approach of Donabedian is valuable in conceptualizing and organizing medical quality evaluation,^{Error! Reference source not found.} and has been widely adopted in the research of ERAS quality evaluation.^{Error! Reference source not found.} In this qualitative interview study, the SPO model was used to analyze the barriers of structure, process, and outcome that potentially affect the implementation of ERAS programs.

4. Key refs on implementation and hurdles for implementation are missing in the ref list and in the discussion.

Response: Thank you for such detailed advice. According to your suggestion, we have added key references on implementation and hurdles for implementation. They are the highlights in lines 558-570, 574-577, 587-590, 595-610, 613-627 and 631-634 of the attached "Main Document-marked copy".

5. Selection of MDT members should be clarified: How many were eligible according to inclusion criteria in the different institutions, how many were not included (reasons)?

Response: Thank you for such detailed advice. We have added this part to the sample of the method. They are the highlights in lines 166-168 of the attached "Main Document-marked copy". The specific content is as follows:

42 members were included in the analysis, including 11 surgeons, 10 anaesthesiologists, 14 surgical nurses and 7 dietitians; 9 were excluded: 2 for poor quality of recording, 2 for schedule clash, 5 for data saturation.

6. To what extent can the results be generalized to the world or at least to the rest of China?

Response: Thank you for such detailed advice. According to your suggestion, we described our expected influence in China. They are the highlights in lines 481-483 of the attached "Main Document-marked copy". The specific content is as follows:

This study is a multicentre qualitative study, and all hospitals included belong to the southern region of China. Therefore, our results can partially reflect the application status of ERAS in the southern region.

7. some details on ERAS implementation should be given: implementation process, year of implementation in the centers, fields of surgery implemented; provide institutional protocol as online appendix; provide some data on compliance in the different centers.

Response: Thank you for such detailed advice. According to your suggestion, we have uploaded supplementary material including the implementation process, year, surgical field, compliance, and institutional protocol. The institutional protocol we signed in September is the Chinese version. If you need the English version, we will re-sign the institutional protocol later.

8. The discussion is full of repetition of results: please moit redundancy but set your results in the context of the literature and draw conclusions how this all will change your pratice or shape new research projects?!

Response: Thank you for such detailed advice. According to your suggestion, we have rewritten the content of the discussion. They are the highlights in lines 370-379, 386-390, 404-410, 413-414 and 430-439 of the attached "Main Document-marked copy".

9. Please provide tables in the end of the manuscript and not embedded in the text.

Response: Thank you for such detailed advice. According to your suggestion, we have made changes and put tables at the end of the manuscript.

10. Table 1 could be nicely provided as figure.

Response: Thank you for such detailed advice. According to your suggestion, we converted the table into a figure, named Figure 1.

Reviewer: 2

Dr. Leah Gramlich , University of Alberta

1. The rationale for using the donabedian approach - SPO - needs more justification and perhaps an acknowledgement of other potential approaches.

Response: Thank you for such detailed advice. According to your suggestion, we have added the content of the theoretical framework to the method. This part of the content introduces in detail what is the SPO model, the specific content of the SPO model, the reason why we choose the SPO model, and how SPO guides our research. They are the highlights in lines 136-147 of the attached "Main Document-marked copy ". The specific content is as follows:

The SPO model proposed by Donabedian considers both internal and external organizational factors. Internally, it covers an organization's structure (S), process (P), and outcome (O), and their interactions. On the other hand, external factors, such as health and social factors are also combined in this model.^{Error! Reference source not found. Error! Reference source not found.} The SPO model also has theoretical underpinnings in that good structure should promote good process and good process should promote good outcomes, which is conducive to guiding the effective improvement of medical quality.^{Error! Reference source not found.} Therefore, the approach of Donabedian is valuable in conceptualizing and organizing medical quality evaluation,^{Error! Reference source not found.} and has been widely adopted in the research of ERAS quality evaluation.^{Error! Reference source not found.} In this qualitative interview study, the SPO model was used to analyze the barriers of structure, process, and outcome that potentially affect the implementation of ERAS programs.

2. The lack of description of ERAS program characteristics/expectations within the programs (eg - do they use an evidence based care pathway, do they use implementation teams to review data and practices with change management approaches, do they use an audit system to inform practice change) is a gap.

Response: Thank you for such detailed advice. According to your suggestion, we have added the description of ERAS program characteristics/expectations within the programs. They are the highlights in lines 370-373 of the attached "Main Document-marked copy ". The specific content is as follows:

Although medical staff implement the ERAS programs under the guidance of the ERAS guidelines, we found that the audit system has not been established or perfected, many feedback problems

cannot be effectively resolved, causing MDT members to face many barriers in the process of implementing ERAS.

3. The authors do not describe involvement of either patients or health system leaders - this is a gap that could be explained in the discussion.

Response: Thank you for such detailed advice. According to your suggestion, we have provided supplementary explanations for this part of the content. They are the highlights in lines 468-473 of the attached "Main Document-marked copy". The specific content is as follows:

Finally, this study did not interview members of all disciplines involved in the implementation of ERAS, such as pharmacists and rehabilitation doctors. In addition, the attitudes and opinions of health system leaders and patients play a vital role in the application and development of ERAS. In the future, we will further expand the types of research objects and improve the research content.

4. The current evidence on barriers and enablers to ERAS care is not completely described based on the literature.

Response: Thank you for such detailed advice. According to your suggestion, we have added key references on barriers and enablers to ERAS care. They are the highlights in lines 558-570, 574-577, 587-590, 595-610, 613-627 and 631-634 of the attached "Main Document-marked copy".

5. The process/approach to qualitative evaluation is not well described - eg how and whata themes were identified. this needs a significant rewrite but it is important information.

Response: Thank you for such detailed advice. According to your suggestion, we described the collection and analysis of the data in the method in detail. They are the highlights in lines 180-235 of the attached "Main Document-marked copy". The specific content is as follows:

Data collection

Interviewees were categorized by location of work site (i.e., Sichuan, Jiangsu or Guangxi Province) and role (i.e., surgeon, anaesthesiologists, surgical nurse and dietitian). The participants were interviewed individually in the period from September to December 2020. Participants in Chengdu, Sichuan Province, conducted face-to-face interviews in a quiet room at their hospitals. Participants from other provinces were interviewed by telephone. The principal investigator (Dan Wang) and research assistant (Jing Zhou, Xinrong Chen and Chengting Chang) conducted the interviews together. All interviewers has completed training in interview. None of the participants had a close

personal relationship with the interviewers, and all participants agreed to be audiotaped. According to the SPO model, a semi-structured topic guide was used for the interviews, as follows:

- ▶ Judging from the current hospital policy management and basic resources, what do you think are the key factors that hinder the implementation of ERAS?
- ▶ During the perioperative period, which measures recommended by the ERAS guidelines have barriers and difficulties in implementation? What factors caused these barriers?
- ▶ What difficulties are faced by multidisciplinary teams in the process of cooperating to implement ERAS programs?
- ▶ What barriers do patients and their families cause to the implementation of ERAS programs?
- ▶ From your professional perspective, do you think what measures should be taken to improve the implementation effect of ERAS programs?

During the interview, we flexibly asked different questions based on the respondent's answers and the semi-structured nature of the interviews allowed participants to explore other topics which they considered relevant. Meanwhile, interviewers promptly confirmed the vague information provided by the interviewees. The average interview time was 21.84 minutes. At the end of interviews, the researcher made a brief summary to determine whether there was any missing or additional information and asked the interviewees to fill in the general situation form. The researcher undertook reflection after each interview to check whether the interview process was inappropriate or needed to be improved and to confirm whether the data had reached saturation and whether the interview process needed to continue. This work adheres to Standards for Reporting Qualitative Research (SRQR).

Data analysis

The recorded interview was transcribed verbatim within 24 hours by member of the research team. Transcribed scripts were assigned a unique code in the order of interviews, and names were removed to align with confidentiality. Data analysis followed an thematic approach of induction and explanation,^{Error! Reference source not found.} applying principles of constant comparison to analyze differences across cases.^{Error! Reference source not found.} Main concepts and themes within the data were identified through a combination of open coding and thematic analysis.^{Error! Reference source not found.} ^{Error! Reference source not found.} The research team repeated line-by-line reading of the transcripts. Following familiarization, the research team manually did open coding, drawing on their rich clinical experiences, and recorded these codes on the margin of the printed transcript.^{Error! Reference source not found.} These codes were subsequently grouped according to the SPO model. Through clustering and integration of codes, themes of each dimension were determined. Themes were considered for inclusion in this report if they were prominent. Prominent themes were further explored in context through in-depth reading and refined until saturation was reached.^{Error! Reference source not found.} To ensure reliability, the research team met regularly to review the coded data, verify its relevance to main themes, discuss the

interpretations and agree on any new theme which were required. Data collection ceased when no new codes or themes were identified, which meant data saturation. Data saturation was considered to be the point at which coded data from new interviews were only added to existing themes and no new themes were developed.^{Error! Reference source not found.} This study finally identified eight themes.

VERSION 2 – REVIEW

REVIEWER	HUBNER, Martin CHUV LAusanne
REVIEW RETURNED	09-Jan-2022
GENERAL COMMENTS	The authors provided a very careful revision of their interesting and important manuscript which delivers an important information presented in a clear and succinct manner: comngratulations!